# Embodied Laser Attack: Leveraging Scene Priors to Achieve Agent-based Robust Non-contact Attacks

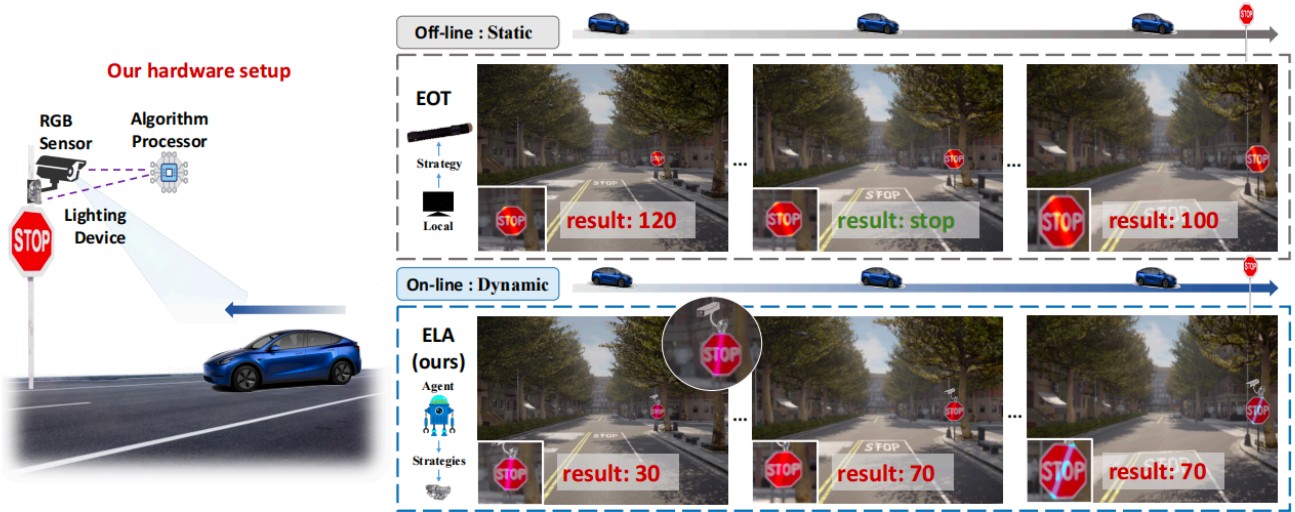

Figure 1: Demonstration of non-contact attack combined with EOT and ELA. **Right:** EOT aims to get a static universal adversarial laser for all scenarios during the sample generation phase before attacks, while ELA trains an agent to make real-time decisions according to scenario changes in a dynamic manner. **Left:** An illustration of our ELA hardware, including an RGB sensor to capture the victim's driving state, an algorithm processor to decide attack strategies, and lighting equipment to conduct. These hardware are installed in suitable locations according to the scene.

## ABSTRACT

As physical adversarial attacks become extensively applied in unearthing the potential risk of security-critical scenarios, especially in dynamic scenarios, their vulnerability to environmental variations has also been brought to light. The non-robust nature of physical adversarial attack methods brings less-than-stable performance consequently. Although methods such as Expectation over Transformation (EOT) have enhanced the robustness of traditional contact attacks like adversarial patches, they fall short in practicality and concealment within dynamic environments such as traffic scenarios. Meanwhile, non-contact laser attacks, while offering enhanced adaptability, face constraints due to a limited optimization space for their attributes, rendering EOT less effective. This limitation underscores the necessity for developing a new strategy to augment the robustness of such practices. To address these issues, this paper introduces the Embodied Laser Attack (ELA), a novel framework that leverages the embodied intelligence paradigm

of Perception-Decision-Control to dynamically tailor non-contact laser attacks. For the perception module, given the challenge of simulating the victim's view by full-image transformation, ELA has innovatively developed a local perspective transformation network, based on the intrinsic prior knowledge of traffic scenes and enables effective and efficient estimation. For the decision and control module, ELA trains an attack agent with data-driven reinforcement learning instead of adopting time-consuming heuristic algorithms, making it capable of instantaneously determining a valid attack strategy with the perceived information by well-designed rewards, which is then conducted by a controllable laser emitter. Experimentally, we apply our framework to diverse traffic scenarios both in the digital and physical world, verifying the effectiveness of our method under dynamic successive scenes.

## CCS CONCEPTS

• **Computing methodologies → Scene understanding**; **Object recognition**.

## KEYWORDS

Dynamic Robustness, Non-contact Attack, Embodied Intelligence

## 1 INTRODUCTION

In many vision-critical applications [2, 27, 30], the perception system heavily relies on deep neural networks (DNNs) to sense and

*ACM MM, 2024, Melbourne, Australia*
© 2024 Copyright held by the owner/author(s). Publication rights licensed to ACM.
ACM ISBN 978-x-xxxx-xxxx-x/YY/MM
https://doi.org/10.1145/nnnnnnn.nnnnnnn

analyze external information, and the security of these intelligent systems is vital. However, since Kurakin et al. [15] propose achievable physical adversarial samples, the increasing diversity of practical attacks [6, 14, 20, 34, 35, 39] have posed a significant threat to the real-life DNNs' deployment, which deserves further exploration to pave the way for enhanced reliability. Notably, their performances are unstable, which can be easily affected by various real-world factors, especially the changes in the target's viewpoint, distances, etc. that often occur in dynamic continuous processes such as traffic scenarios. Thus, an urgent need exists to enhance the practical attacks' robustness when faced with such possible variations in the real world to obtain consistently stable performance.

To enhance the robustness of attack methods faced with dynamic variations, Expectation over Transformation (EOT), introduced by Athalye et al. [1], plays a crucial role, particularly in optimizing contact attacks like adversarial patches. These patches, while robust and universally applicable due to EOT optimization, suffer from a lack of concealment and practical challenges associated with their physical placement in the real world. Conversely, non-contact optical attacks, leveraging light manipulations, i.e. manipulating the slope, width, and wavelength of a laser, provide greater stealth and are easier to deploy with no need for manual attachment. However, the limited variability in light parameters makes it difficult for EOT to effectively develop a universal strategy that can adapt to changing conditions with a simple style.

Given the inherent challenges of enhancing such attacks in dynamic scenarios, this paper aims to focus on non-contact attacks that are harder to boost performance and explores a new method from the perspective of active adaptation. Specifically, We introduce a dynamic, robust laser attack framework, the "Embodied Laser Attack (ELA)", applied here within a traffic scenario as a case study. The ELA framework leverages the core concepts of embodied intelligence—"Perception-Decision-Control"—to dynamically adjust a highly manipulable laser medium, crafting optimal real-time attack strategies based on current perceived states. This method represents a departure from traditional static, offline enhancements such as EOT, focusing instead on active adaptation to enhance the robustness of non-contact attacks. A visual comparison between EOT and our ELA method is illustrated in Figure 1.

However, to implement this idea, there exist two challenges: **(1) Perception Transformation for Victim's Perspective.** In practical scenarios, acquiring information from the victim is challenging, while data from a third-party sensor is more obtainable, meanwhile posing a necessity of transforming data from the attacker's view to the victim's. Moreover, unlike static attacks focusing on a singular scene, dynamic scenarios present both the attack target and surrounding environment as ever-changing, requiring the real-time and precise acquisition of target information. Therefore, our first challenge is to efficiently and rapidly utilize such accessible data to establish real-time inference between the two perspectives, on which attack strategies depend. **(2) Dynamic Decision with Flexible Control.** For the decision and control (DC) stage, we need to interactively learn from the dynamic scenario and achieve a rapid response with a flexible way to conduct as well. Existing non-contact optical attacks in traffic scenarios often fail to provide instantaneity and efficiency due to their reliance on query-based algorithms that are time-consuming. So, how to utilize perceived

information to construct and launch instantaneous attacks is our second challenge.

To address the above issues, firstly, we innovatively utilize the intrinsic geometric priors of traffic scenes, which significantly reduce the computational cost associated with traditional full-image perspective transformation technologies. Specifically, a few-layer Multilayer Perceptron (MLP) is utilized to construct the Perspective Transformation Network (PTN), which could enable a real-time and efficient estimation of the target's distorted states in the victim's view from the attacker's accessible imaging. Secondly, we propose agent-based decision-making and manipulate the laser (shooting angles, wavelength, etc) — a flexible and controllable medium to implement in the physical world for the DC module, which trains an attack agent with reinforcement learning, capable of instantaneously determining a valid strategy based on the perceived status estimation during the attack phase. Besides, we carefully design the reward function to further guarantee the immediacy of attacks and the laser's physical properties. Main contributions are as follows:

- We propose the **first dynamic robust non-contact attack** framework called ELA, which utilizes the paradigm of embodied intelligence: Perception-Decision-Control to dynamically and timely adjust a manipulable laser towards valid attack strategies according to current situations, rather than statically enhancing a fixed physical adversarial example's robustness in an off-line manner like EOT ahead of time.
- We address two challenges in ELA: A novel Perspective Transformation Network (PTN) is proposed to enable rapid simulation of object variations relying on the intrinsic geometric priors of traffic scenes; An adversarial laser decision-making agent is designed to facilitate real-time valid strategies for dynamic successive processes.
- We evaluate our method on the sign recognition task in traffic scenarios based on data collected from the CARLA simulator and the real world. Experimental results demonstrate the effectiveness of our framework for different categories and complex scenes, including various viewpoints, distances, etc.

## 2 RELATED WORKS

### 2.1 Robust Physical Adversarial Attacks

Considering that physical attacks [4, 5, 9, 22] are vulnerable to changeable environmental conditions, several works have been proposed to ensure the robustness of attack methods in the real world. Among them, EOT [1] is a classic technique utilized in the majority of existing physical attacks, which applies various random transformations, such as scaling, rotation, and blurring, to a single sample, thereby improving the universality of adversarial examples across environmental variations. In addition, there also exist some other works, like Robust Physical Perturbations (RP2) [9] by Eykholt et al. that create visually deceptive modifications to objects like road signs, Xu et al. [37]'s Adversarial T-shirt that models non-rigid deformation to enhance the robustness towards human movement. These advancements indicate a growing sophistication in adversarial techniques, focusing on practical challenges such as dynamic environments and deformable objects, thereby pushing the boundaries of physical-world adversarial research.

Figure 2: An overview of the ELA framework, which consists of two main modules for robust laser attacks. The attacker first captures the vehicle through a fixed sensor, and the perception module infers the object's state (location, scale, and distortion) in the victim's view based on the characteristic of shape change. Then the DC module utilizes such region information to autonomously make real-time decisions and controls the light projection hardware to achieve continuous physical attacks.

However, these robustness enhancement techniques, predominantly designed for contact attacks, tend to be less effective when applied to non-contact adversarial attacks such as those involving lasers or other remote manipulation methods. Non-contact attacks typically use mediums like lasers, which are inherently homogenous compared to textured patches, making techniques like EOT difficult to find a unified attack strategy within such a limited parameter space for optimization. Therefore, despite non-contact attacks' flexibility and concealment, their robustness is often poor. To bridge this gap, our work shifts focus towards enhancing the robustness of non-contact adversarial attacks through an active adaptation framework rather than static enhancement like EOT.

## 2.2 Attacks in Traffic Scenarios

It should be noted that most robust physical attack methods need to be manually attached to the target, which is not always practical, and it is difficult to adapt to other situations once implemented, thus causing limitations in traffic scenarios. To address this issue, some achievable non-contact attacks in the physical world are designed with adversarial light as an easy-to-control medium, such as [12, 18] using a projector to attack recognition tasks with a designed texture projection, and [8, 11, 38] conducting attacks with geometric light on the whole image to mislead traffic sign recognition. In our method, we also utilize light due to its flexibility, specifically adding a laser beam on the traffic sign to promise being wholly captured by the lens, which is easy to conduct.

Notably, we find that most light attacks adopt real-time optimization using evolutionary algorithms like Genetic algorithm (GA) [11] and Particle Swarm Optimization algorithm (PSO) [29, 40], or greedy algorithm [8, 16], which will lead to uncontrollable performance and time costs and hinder those applications in dynamic scenarios. Unlike those methods, we attempt to design a different optimization method with reinforcement learning, ensuring its attack performance and speed simultaneously.

## 3 METHODOLOGY

In this section, we introduce our proposed Embodied Laser Attack (ELA) based on the safety-critical task: traffic sign recognition. The whole framework is presented in Figure 2.

## 3.1 Problem Definition

The inherent challenge in non-contactly attacking traffic sign recognition tasks lies in dynamic scenarios. The variability and unpredictability of real-world conditions range from changes in alterations in the vehicle's speed to the angle of view, which demand an adaptive attack method that can consistently fool the recognition systems under such fluctuating circumstances.

To address the identified challenge, we propose a novel attack framework **ELA**, which effectively integrates the paradigm of embodied intelligence: **perception-decision-control**. Specifically, our ELA is first meticulously designed to enable the active perception of real-time, precise data. This process involves a critical perspective transformation $\mathcal{T}$ that could transform the attacker view $\mathbf{o}_a^t$ observed by a third-party sensor into the victim view $\mathbf{o}_v^t$. Then, we combine $\mathbf{o}_v^t$ with light parameters $\mathbf{l}_\theta^t$ into the state $\mathbf{s}_t$ to an agent developed within a reinforcement learning paradigm, enabling it to make an instantaneous decision $\mathbf{a}_t$ corresponding to each state $\mathbf{s}_t$. Additionally, the chosen medium for this agent's interaction with the environment is laser beam, selected for its inherent controllability, and the action space of the agent is represented by the laser's several attributes.

Based on the above, the objective of optimization within our ELA framework can be succinctly summarized as follows:

$$\max_{\phi,\theta} \sum_{t=1,\cdots,T} \mathbb{I}(\overline{\mathbf{y}}_t \neq \mathbf{y}_t), \qquad (1)$$

with $\overline{\mathbf{y}}_t = f\left(\mathcal{A}(\mathbf{s}_t, \mathbf{a}_t)\right)$, $\mathbf{s}_t = \{(\mathcal{T}(\mathbf{o}_a^t; \phi_\mathcal{T}), \mathbf{l}_\theta^t\}, \mathbf{a}_t \sim \pi(\cdot|\mathbf{s}_t; \Theta)$,

where $\mathbb{I}(\cdot)$ signifies an indicator function that equals 1 when $\overline{\mathbf{y}}_t \neq \mathbf{y}_t$, $\phi_\mathcal{T}$ and $\Theta$ respectively represent parameters for the perspective transformation $\mathcal{T}$ and the decision-making policy $\pi$. $\mathbf{s}_t$ is the actual state with an estimation of the victim's perspective from $\mathcal{T}$ as well as $\mathbf{l}_\theta^t$. $\mathcal{A}(\cdot)$ represents an operation that performs the laser attack strategy determined by $\mathbf{a}_t$, which is chosen by $\pi$ according to current state $\mathbf{s}_t$, to generate the final adversarial image for recognition through the classifier $f$. In the following sections, we will detail the core designs in the perception module and the DC module.

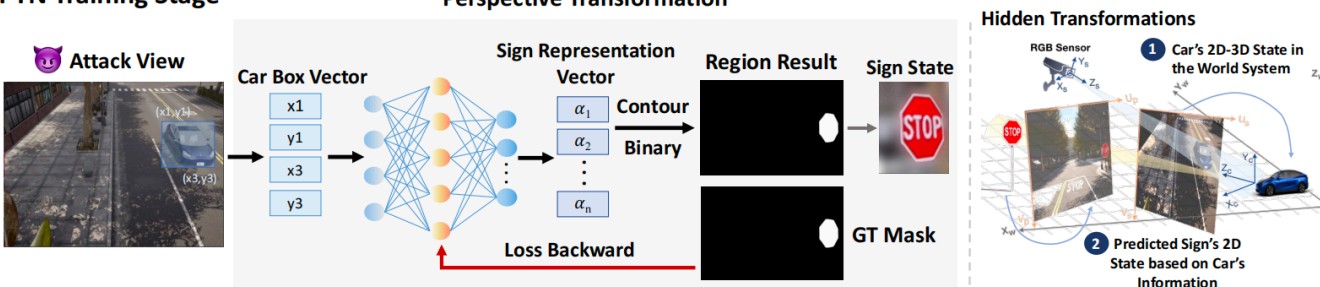

**Figure 3: Left: The training and inference process of PTN. By leveraging the vehicle's position in sensor imaging, PTN could simulate the target's region in the victim's view by spatial transformation. Right: Derivation process executed by PTN.**

## 3.2 Perception Module

Typically, in the actual attack scenarios, directly accessing data from the victim's perspective is impractical. To address this, we propose a third-party perception module (shown in Figure 3), grounded in the concept that acquiring third-party scene data via an externally placed sensor is more feasible. However, it's obvious that the data acquired from such a third-party standpoint does not align with the imaging observed from the victim.

To address the inherent imaging inconsistencies at different viewpoints, traditional approaches often rely on multi-view imagery for new viewpoint synthesis, which suffers from high time costs and loss of actual precision. Given that traffic sign recognition in real scenarios involves critical region localization before classification, slight position deviations in generated viewpoint information may hinder the attack performances drastically. Compared to such full-image perspective simulations, we propose a local transformation method by leveraging traffic-specific shape priors, significantly enhancing accuracy and efficiency in perspective estimation.

*3.2.1 Prior Knowledge in Traffic Scenarios.* Traffic scenarios possess unique prior knowledge, inherently existent in both the third-party attacker and the victim's perspective, such as geometric shapes. For instance, vehicles in the scene from the attacker's view can be represented by rectangles, while traffic signs in the scene from the victim's view often adhere to fixed shapes as well, such as circles or octagons. For traffic sign recognition, what is essential is merely the portion of the victim's viewpoint containing the traffic sign. Therefore, we can abstract complex scenes from pixel representations to representations based on shape priors.

For the scene $\mathbf{o}_a^t$ under the attacker view, as demonstrated in Figure 3, we can represent $\mathbf{o}_a^t$ into the minimal bounding box wrapping tightly around the vehicle's contour in the imaging, a rectangle whose shape depends on the vehicle's location and rotation. This representation captures the scene's essence by focusing on the geometric stability of vehicles, a persistent element in traffic scenarios. Mathematically, $\mathbf{o}_a^t$ could be formulated as follows:

$$\mathbf{o}_a^t = [\mathbf{x}_{min}^t, \mathbf{y}_{min}^t, \mathbf{x}_{max}^t, \mathbf{y}_{max}^t], \tag{2}$$

where $\mathbf{x}_{min}^t, \mathbf{y}_{min}^t$ denotes the coordinates of the top-left vertex, and $\mathbf{x}_{max}^t, \mathbf{y}_{max}^t$ represents the coordinates of the bottom-right vertex of the minimal bounding rectangle at time $\mathbf{t}$.

For the scene $\mathbf{o}_v^t$ under the victim view, it can similarly be characterized using the constant presence of the traffic sign. Specifically, with distinct shapes such as circular or octagonal, we can represent each using the coordinate of its center position along with a geometric parameter set $\boldsymbol{\eta}_{geo}^t$. Thus, $\mathbf{o}_v^t$ can be expressed as:

$$\mathbf{o}_v^t = [\mathbf{x}_{center}^t, \mathbf{y}_{center}^t, \boldsymbol{\eta}_{geo}^t], \tag{3}$$

where $(\mathbf{x}_{center}^t, \mathbf{y}_{center}^t)$ denotes the coordinate of the sign's center at time $\mathbf{t}$, and $\boldsymbol{\eta}_{geo}$ can be further reduced to its geometric shape in the victim's view, for example, a circle sign 30's $\boldsymbol{\eta}_{geo}^t$ consists of the respective lengths $a, b$ of an ellipse's long and short axes as well as the angle $\Delta$ of deflection to characterize the distorted circle, i.e. $\boldsymbol{\eta}_{geo}^t = [a, b, \Delta]$.

*3.2.2 Perspective Transformation with Scene-prior Knowledge.* After obtaining scene representations $\mathbf{o}_v^t, \mathbf{o}_a^t$ based on scene-prior knowledge, we need to consider how to conduct a transformation from $\mathbf{o}_a^t$ to $\mathbf{o}_v^t$. Drawing inspiration from pioneering research [17, 19] that exploits the potent modeling prowess of multi-layer perceptrons (MLP), we have conceived and implemented a learning-based Perspective Transformation Network (PTN) tailored to our specific requirements, as illustrated in Fig 3. Thus, the transformation process $\mathcal{T}$ can be expressed as follows:

$$\mathbf{o}_v^t = \mathcal{T}(\mathbf{o}_a^t; \phi_{\mathcal{T}}), \tag{4}$$

where $\phi_{\mathcal{T}}$ represents the learnable parameters of PTN.

Regarding the training of this network, once the state $\mathbf{o}_v^t$ is obtained, we employ geometric knowledge to further locate the traffic sign. Taking a circular sign 30 as an example, we define the sign's religion based on the sign's geometric parameters $\boldsymbol{\eta}_{geo}^t = [a, b, \Delta]$. This process allows us to generate a mask that precisely indicates the traffic sign's region. For example, the creation of a mask $\mathbf{M}$ for the sign area, based on the geometric parameters $\boldsymbol{\eta}_{geo}^t$, can be formalized as follows:

$$\mathbf{M}(i, j) = \begin{cases} 1 & \text{if } \frac{((i-\mathbf{x}_{center})\cos(\Delta)-(j-\mathbf{y}_{center})\sin(\Delta))^2}{a^2} \\ & + \frac{((i-\mathbf{x}_{center})\sin(\Delta)+(j-\mathbf{y}_{center})\cos(\Delta))^2}{b^2} \leq 1 \\ 0 & \text{otherwise} \end{cases} \tag{5}$$

where $(i, j)$ represents the coordinates of pixel in the mask $\mathbf{M}$.

Then, upon obtaining the predicted mask $\mathbf{M}$, it can be compared with the ground truth mask on a pixel-by-pixel basis to calculate the

Mean Squared Error (MSE) loss. This computed loss is then utilized to iteratively update the PTN. The loss function is as follows:

$$\mathbf{L}_{\text{MSE}} = \frac{1}{NK} \sum_{i=1}^{N} \sum_{j=1}^{K} (\mathbf{M}(i, j) - \mathbf{M}_{\text{GT}}(i, j))^2 \qquad (6)$$

where $N, K$ denote the number of rows and columns in the mask matrix, $\mathbf{M}(i, j)$ is the value of the predicted mask at pixel $(i, j)$, and $\mathbf{M}_{\text{GT}}(i, j)$ is the value of the ground truth mask at the same pixel.

As the PTN converges through the training process, it becomes good at transforming perspectives from the attacker's to the victim's viewpoint across various scenes. This capability is derived from the PTN's focus on inferring the location mask without the need to consider the specific content within the scene, where the simulation of the target sign generally aligns with the actual distortion state of the target. Therefore, during the whole moving process, PTN can generalize the attacker's perspective information shift to the victim's viewpoint.

## 3.3 Decision and Control Module

After estimating the scenario $\mathbf{o}_v^t$ from the victim's perspective, there remains a need for an instantaneous attack method. Consequently, we have developed a novel agent-based attack framework, which is particularly well-suited to autonomous and timely decisions in dynamic scenarios. Detailed explanations will follow.

*3.3.1 Basic Definition.* Regarding our agent-based attack framework tailored for dynamic environments, we first focus on basic components in the RL agent $\mathbf{A}$, i.e. states $\mathbf{s}_{t=1,\cdots,T}$ and action space $\mathcal{U}$ of $\mathbf{A}$. As mentioned in Eq. (3), the victim view $\mathbf{o}_v^t$ at time $\mathbf{t}$ can be transformed from the attacker view $\mathbf{o}_a^t$ by PTN $\mathcal{T}(\cdot; \phi_{\mathcal{T}})$. Thus, combining the scene $\mathbf{o}_v^t$ under the victim view with the parameter $\mathbf{l}_\theta^t$ of the laser beam to be applied in the scene, we can characterize the state $\mathbf{s}_t$ at this time as follows:

$$\mathbf{s}_t = \{\mathbf{o}_v^t \leftarrow \mathcal{T}(\mathbf{o}_a^t; \phi), \mathbf{l}_\theta^t\}, \quad \text{with } \mathbf{l}_\theta^t = \{\mathbf{k}_t, \boldsymbol{\omega}_t, \boldsymbol{\lambda}_t\}, \qquad (7)$$

where $\mathbf{k}_t, \boldsymbol{\omega}_t, \boldsymbol{\lambda}_t$ denote the slope, width, and wavelength of the laser beam at time $\mathbf{t}$, respectively.

Given our selection of the laser beam as the control medium, the action space $\mathbb{A}$ of the agent $\mathbf{A}$ is defined by the parameters $\mathbf{l}_\theta$ of the light encoded within the state representation. Mathematically, the action space $\mathbb{A}$ can be characterized as follows:

$$\mathbb{A} = \{(\mathbf{k}, \boldsymbol{\omega}, \boldsymbol{\lambda} \mid \mathbf{k} \in \mathbb{R}, \boldsymbol{\omega} \in \mathbb{R}^+, \boldsymbol{\lambda} \in [\lambda_{\min}, \lambda_{\max}]\}, \qquad (8)$$

where the range $\lambda_{\min} = 400$ nm $\sim \lambda_{\max} = 700$ nm encompasses all the colors that the human eye can perceive. Next, we will discuss how to train policy network $\pi$.

*3.3.2 Training Stage of Policy Network.* To ensure the adaptability of the adversarial laser beam across consecutive time frames, and different driving environments, our training dataset is collected from various driving scenes and pre-processed into distinct, continuous five-frame segments. The agent is trained on these segments via random sampling to enhance its ability to generalize across dynamically changing scenarios. Then, given the continuous nature of the light parameter space, and to enable more effective exploration and adaptation, our optimization framework employs the Proximal Policy Optimization (PPO) algorithm, as detailed in [21]. This method leverages a modified objective function that facilitates

stable and effective policy updates by utilizing the probability ratio and advantage estimates. The training objective of PPO is designed to minimize the expected cost, formalized as:

$$\min_{\Theta} \mathbb{E}_{t,\tau} \left[ -\min \left( r_t(\Theta) A_{t,\tau}, \text{clip}(r_t(\Theta), 1 - \epsilon, 1 + \epsilon) A_{t,\tau} \right) \right], \quad (9)$$

where $\Theta$ denotes the parameters of the policy $\pi$. The term $\mathbf{r}_t(\Theta) = \frac{\pi(\mathbf{a}_t | \mathbf{s}_t; \Theta)}{\pi(\mathbf{a}_t | \mathbf{s}_t; \Theta_{old})}$ represents the probability ratio, comparing the likelihood of selecting action $\mathbf{a}_t$ under the current policy with that under the previous policy. $A_{t,\tau}$ denotes the advantage function at time $t$ within the training segment $\tau$, measuring the relative benefit of the chosen action over the expected value of all possible actions in that state. $\epsilon$ is a critical hyperparameter that controls the clipping range, ensuring that updates are appropriately scaled to prevent destabilizing the learning process or stalling improvements.

Because the relationship between the advantage function $A_t$ and the reward function $\mathcal{R}$ in PPO is well-understood and straightforward, here we directly introduce the design of reward functions. Specifically, we first design a reward $R_{\text{attack}}$ for the efficiency of the attack and its time cost. It should be noted that since direct interaction with the victim model is impractical, we utilize ensemble surrogate models $F_{1,\cdots,n}$ to evaluate the effectiveness of our strategies. The formulation of $R_{\text{attack}}$ is represented as follows:

$$\mathcal{R}_{\text{attack}} = \begin{cases} \mathbf{r}_s - \alpha \cdot N_{\text{steps}}, & \text{if success} \\ -\frac{1}{n} \sum_{i=1}^{n} c_i \cdot F_i - \alpha \cdot N_{\text{steps}}, & \text{otherwise} \end{cases}, \quad (10)$$

where $\mathbf{r}_s$ is the reward for a successful attack. $F_i$ represents the confidence score from the $i$-th model in the ensemble. $c_i$ is the scaling factor for the $i$-th model's contribution to the penalty. $\alpha$ is a penalty factor for the number of steps $N_{\text{steps}}$, which encourages the development of more efficient strategies.

Besides, to generate a valid light with the principle of keeping the disturbance as little as possible, we need to limit the area covered by the light, which mainly depends on its width, and the wavelength should also be proper or it cannot be captured by the victim's sensor. Based on the above requirements, we design another reward $\mathcal{R}_{\text{appear}}$ for the natural appearance of lights:

$$\mathcal{R}_{\text{appear}} = (\boldsymbol{\omega}_0 - \boldsymbol{\omega}) \cdot \mathbf{r}_\omega + \mathbb{I}[(\lambda - \lambda_{\min}) \cdot (\lambda_{\max} - \lambda)] \cdot r_\lambda, \quad (11)$$

where $\mathbf{r}_\omega$ represents a predefined reward value for the beam width. When the beam width, denoted as $\boldsymbol{\omega}$, is less than a threshold $\boldsymbol{\omega}_0$, a reward of $(\boldsymbol{\omega}_0 - \boldsymbol{\omega})\mathbf{r}_\omega$ is granted. Conversely, if the beam width exceeds this threshold, it results in a penalty. Similarly, $\mathbf{r}_\lambda$ follows a comparable mechanism. $\lambda_{\min}$ and $\lambda_{\max}$ could be set by 400 and 700 as the range borders, and $\mathbb{I}$ is the signum function.

Finally, those rewards work together as a comprehensive evaluation to train our agent $A$ with guidance:

$$\mathcal{R} = \gamma_1 \mathcal{R}_{\text{attack}} + \gamma_2 \mathcal{R}_{\text{appear}}, \qquad (12)$$

where $\gamma_1$ and $\gamma_2$ are the weights that adjust the importance.

Then, utilizing the trained policy network $\pi$, we can sample action $\mathbf{a}_t$ from $\pi$, to adjust the laser beam parameters $\mathbf{l}_\theta^t$ for generating adversarial examples $\mathcal{A}(\mathbf{s}_t, \mathbf{a}_t)$ at time $\mathbf{t}$ during inference. Additionally, during the inference process, aside from the initial state $\mathbf{s}_0$ where random light parameters are used, the light parameters for subsequent frames are derived by using the parameters from the previous frame as the initial parameters for inference. Compared with [8, 16, 29, 40] treating separate frames equally and

randomly, our agent could cope with the whole consecutive process through interaction and learning, making effective use of scenarios and empirical information, and our attack phase is more efficient.

# 4 EXPERIMENTS

## 4.1 Settings

**Simulation of Physical Implementation:** Due to the inherent risks and complexities of implementing ELA in the real world, we conduct most validations within the near-realistic environment of the CARLA Simulator [7] with reference to other physical attacks in autonomous driving scenarios [3, 24, 25, 31–33, 36]. CARLA is an autonomous driving simulator, owning 3D maps with realistic buildings, traffic signs, dynamic vehicles, diverse weather, and manually placable RGB sensors, which is just right for our needs. Specifically, to consider the variety of scenes, we choose three maps with different settings in CARLA: **a small village embedded in mountains with a special infinite highway, a town with skyscrapers, residential buildings and an ocean promenade, as well as a small rural area with a river and several bridges**. Such selections naturally cover various environmental conditions in terms of sunlight, fog, etc. with four different sign settings by default: Sign "30", Sign "60", Sign "90" and Sign "Stop" since the traffic requirements for these scenarios are diverse. Then we record several aligned video pairs (a total of 3200 frames) for each category as training data. For the data obtained from the fixed sensor, we annotate signs at the pixel level, and for data from the victim's view, we annotate the vehicle at the box level. Besides, each category uses frames of an extra video pair shot for a whole driving route (each 100 frames) as the test data. We carry out attacks in CARLA and record videos of the corresponding attack status of signs, and then evaluate the classification results after attacks in an off-line way.

**Victim Models:** For the DC module, our ensemble surrogate models contain some commonly used kinds including ResNet50 [10], Inception-v3 [26], EfficientNet [28], and we select another three main classifiers to test: ResNet101 [10], DenseNet121 [13] and GoogLeNet-V3 [26]. We first train in GTSRB [23] with officially pre-trained weights, and then finetune the models by our own video frames, which achieves 100% precision on clean test samples.

**Metrics:** We choose a commonly used metric to indicate the overall performance: Attack Successful Rate (ASR). Specifically, ASR shows the attack performance as a ratio of successfully misclassified samples in the test set. In our paper, we record a video about the attack and compute the corresponding ASR for the video's frames.

**Other Implementations:** For the part of active perception, we directly apply our trained weights with the best performances to the perspective transformation work during attacks. For the agent training, we set *epoch* as 100 with sampling operations.

## 4.2 Effectiveness of Active Perception

To show the accuracy of our first-perspective inference, we utilize a common metric in the field of computer vision to conduct an assessment: Intersection over Union (IOU). Specifically, we train PTN and make predictions on test datasets of the four categories, and then calculate the average IOU as their overall performances, which here doesn't merely represent the overlap degree of two areas in 2D images, but the effect after many variations involving rotation,

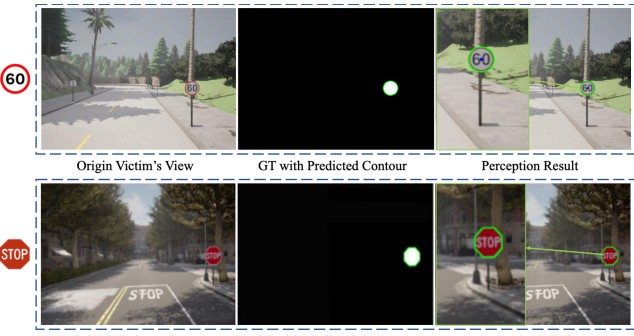

Origin Victim's View    GT with Predicted Contour    Perception Result

**Figure 4: Perception results of targets' states. We choose two examples with different shapes, where the green line in each image is the inference result of our perception module, where we can see that the generated contour coincides well with the actual outline of the ground truth. Such reasoning results could act as a chronological prior insight for decision.**

distortion, and scaling, etc. The results of each category are shown in Tab 1, where we can see that all are above 90%, showing our method's validity. To be clearer, we also give visualized examples of the perception results of two kinds of signs with different shapes: Sign "60" and Sign "Stop" in Fig 4. We can find that the predicted region of the target sign is basically the same as the real imaging situation, which demonstrates the accuracy of our perception module and lays the foundation for effective decision-making on the current state in the next stage.

**Table 1: Perception Performances on four categories.**

| Category | 30 | 60 | 90 | STOP |
|---|---|---|---|---|
| mIOU | 0.9361 | 0.9632 | 0.9145 | 0.9686 |

## 4.3 Effectiveness of DC Module

To demonstrate the effect and rapidity of our decision and control module, we also compare the attack performance with two open-source light attack methods: Adversarial Laser Beam (AdvLB) [8] and Adversarial Laser Spot (AdvLS) [11]. It's clear that they are faced with different application situations: first, the former two methods are black-box static methods designed to search parameters for one scene with no time limit, optimizing through random search and retaining better solutions in an off-line manner, while our method is trained ahead for the whole process to promise real-time and changeable attacks; secondly, their light could appear anywhere on the whole image, while ours is limited by the sign's area. To ensure fairness, we restrict their rays to be only added to the sign area as ours, then we calculate ASR and average time cost (Time) for comparison. Since the above methods use a fixed way to attack, we just calculate the optimization period as a time value, and for our method, we record the time costs of each attack and use the mean of the set as the final result. Here we take attacks on Inception-V3 as an example. The results are shown in Tab 2, where we find that our approach is generally in a good position,

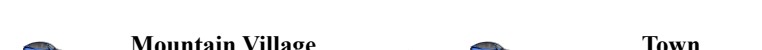

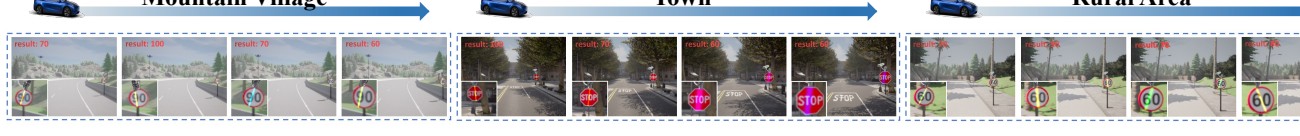

**Figure 5: Examples of ELA's attack for various scenes. Under this framework, we can first accurately estimate the imaging area of the target sign, and our agent can make effective decisions based on perceived information at different moments.**

**Table 2: Attack performances and time costs of the three attacks against the surrogate classifier Inception-V3.**

| Category | 30 | | 60 | | 90 | | STOP | |
|---|---|---|---|---|---|---|---|---|
| Method | ASR↑ | Time↓ | ASR↑ | Time↓ | ASR↑ | Time↓ | ASR↑ | Time↓ |
| AdvLB | 7% | 13.76s | 1% | 18.56s | 15% | 7.10s | 6% | 14.10s |
| AdvLS | 5% | 2.13s | 0% | 2.77s | 17% | 0.46s | 1% | 2.69s |
| **ELA** | **60%** | **0.22s** | **33%** | **0.18s** | **76%** | **0.20s** | **40%** | **0.13s** |

**Table 3: Transfer-based Attack performances of AdvLB [8] under the enhancement of EOT and ELA, respectively.**

| Victim Model | ResNet101 | | DenseNet121 | | GoogLeNet-V3 | |
|---|---|---|---|---|---|---|
| Methods | EOT | **ELA** | EOT | **ELA** | EOT | **ELA** |
| *30 SPEED LIMIT* | 13% | **65%**/+52% | 21% | **73%**/+52% | 46% | **67%**/+21% |
| *60 SPEED LIMIT* | 35% | **63%**/+28% | 9% | **48%**/+39% | 10% | **39%**/+29% |
| *90 SPEED LIMIT* | 28% | **52%**/+24% | 11% | **49%**/+38% | 26% | **45%**/+19% |
| *STOP* | 1% | **39%**/+38% | 0% | **16%**/+16% | 33% | **42%**/+9% |

achieving the best ASR of up to 76% with the highest speed since our agent has learned before, although in some cases the ASR is not particularly high. Since we just use a common light beam without complex texture and attack once, we believe such results are acceptable for this simple pattern. Among them, possessing the same attack form as ours, AdvLB's performance especially highlights the superiority of our attack decision-making, proving that our light attack's significant superiority depends on the decision process, rather than the light design itself which is not our main innovation. Besides, it is worth emphasizing that all methods experience a training phase interaction and cannot receive feedback on the classification information during the one-step attack phase, which is in line with the real application in physical scenarios and means a more challenging attack level.

## 4.4 Systemic Verification for ELA Framework

To conduct systemic verification for our ELA, we have done a comparative experiment with EOT to show the effects of the two different robust attack methods. Considering that EOT and ELA have quite different detail settings, we apply both to our scenario with well-thought-out condition settings for relatively objective verification, and here we both choose to attack three unseen classifiers: ResNet101, DenseNet121, and GoogleNet-V3 to test the transferability of the two methods, simulating real implementation conditions. For EOT, we allow it to use part of the whole training data (1/10) with random variations like random cropping, rotating, blurring,etc, and also choose a laser beam as the attack form. For ours, we remain in the same setting. The results of the four categories against three classifiers are shown in Tab 3, where we can see that our ASR is far higher than Advlb's results, roughly reflecting the effective cooperation between our perception module and the decision module. Obviously, for physical attacks like ray attacks, although they have been improved by the enhancement method EOT, it is still difficult to work for all samples with a unified style like using a complex noise-based patch in the digital world. At this point, our approach is better suited to solve this scenario, which can make changeable decisions with rapid speed, as represented in Fig 5.

## 4.5 Additional Results

**Convergence Discussions:** As an optimization method based on reinforcement learning, it is necessary to explore its convergence to prove its training validity from a quantitative perspective. Therefore, we count the rewards and attack time during the training process and then use line charts to reflect the training trend. Fig 6 demonstrates both the change of rewards and attack time during the training process of the sign "Stop" as an example, where we clearly observe that the rewards have shown a total increasing trend despite relatively obvious volatility. We believe that this change is reasonable since our agent training for each scenario is based on sampling from the whole train dataset, and each frame does not correspond to a unique solution. Meanwhile, it's obvious that when the training epoch arrives at almost 40, the mean optimization step number drops sharply, and then changes always within our step threshold, which could finally promise the attack for each scene to be accomplished timely.

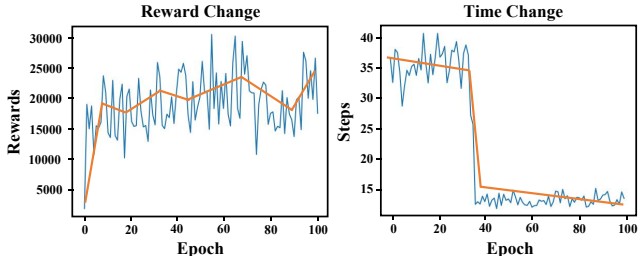

**Figure 6: Demonstration of evolutionary trends of rewards and average attack steps (time) respectively in the training process of sign "Stop".**

**Validity of Agent Training:** To prove the learning capability of our agent, we also compare the results of random attacks with our trained agent-based attacks for different scenarios. The results are shown in Tab 4, where we can clearly see that each category has an extremely significant difference in ASR before and after agent training. Even though the difference in attack difficulty on those

**Pred:** Stop    Stop    Stop    Stop    Stop    Stop    Stop

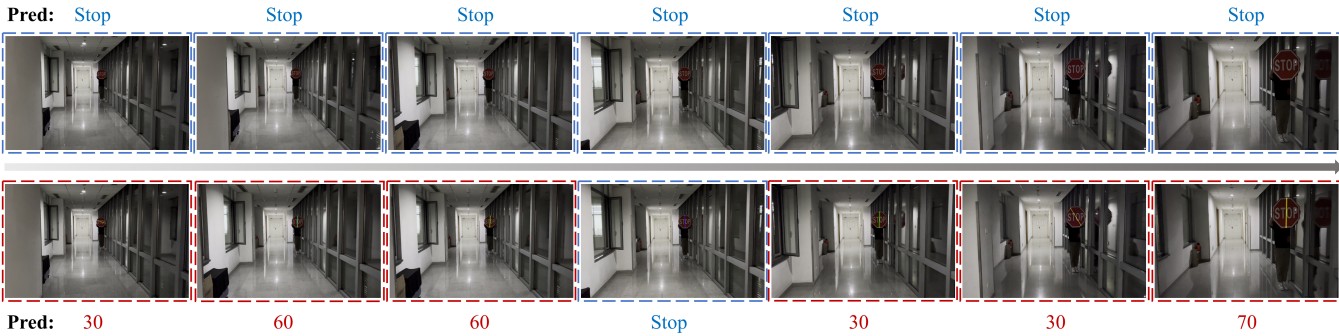

**Pred:** 30    60    60    Stop    30    30    70

**Figure 7: Comparison of partial clean video frames and antagonistic video frames when conducting our laser attack. During such a process, we target the simulation of dynamic driving scenarios in the physical world.**

classifiers is huge, where some agent attacks have relatively worse performance with a gap of almost 30% than the optimal one, our performance is still significantly better than theirs. Such results fully prove that our agent-based decision module can effectively learn and take attack strategies suitable for different situations, and can make timely decisions within the specified time.

**Table 4: Comparison of attack effects before (random) and after agent training in our framework.**

| Category | 30 | 60 | 90 | STOP |
|---|---|---|---|---|
| Random | 10% | 13% | 11% | 0% |
| Agent | **65%**/+55% | **63%**/+50% | **52%**/+41% | **39%**/+39% |

**Verification in the Physical World:** Although we are not at liberty to achieve real-vehicle verification, we still attempt to simulate such a process and validate our method under the approximation of the real-world conditions. In fact, the perspective transformation relations fitted by the perception module are not easily disturbed by other factors in the real world and are sufficiently verifiable in the digital world. Thus, we focus on mimicking a vehicle's real movement. Imitating the real driving situation, one person is asked to stand at a fixed position against the side holding a stop sign, while another person moves forward with a shooting camera, obtaining a video from a pseudo-vehicle imaging perspective. The visual examples of partial frames are listed in the first row of Fig. 7. It is clear that the sign is scaling and shifted with slight deformation during the successive process, which is consistent with real scenarios. We shoot for 7 seconds at twenty frames a second (total 140 frames) within an indoor scene and perform our attack on this video, which finally obtains an average ASR of 31.3% on the three classifiers, the result suggests that the external environment has a certain impact on the performance but it's not particularly significant compared with digital results. We also consider it acceptable since the sign "STOP" has unique discriminatory characteristics and is more difficult to attack than most categories. The attack results of partial frames are listed in the second row of Fig. 7, where we can find that the multi-frame training strategy brings part of the similarity to light patterns across frames.

**Statistical Insights on Attack Results:** We consider that each category has its own representative characteristics as well as a

corresponding misleading tendency, so we discuss the effect of our ELA attack from a macro point of view, globally analyzing the statistical results of the false determinations triggered by the attack and revealing potential risk warning insights. Specifically, we count the frequencies of labels that are incorrectly predicted after successful attacks and attempt to find out such rules with the results in Tab 5, where we can observe that a speed limit sign "30" with an adversarial light can be mostly mistaken for sign "70" or less frequent "60" with higher restrictions, and the action-determining sign "Stop" could also be vulnerable to be misled as others. Based on this, we can easily recognize what weak points the embodied agents tend to dig out for each category, and then future work could focus more on the top frequent labels to ensure reliability.

**Table 5: Statistical findings on the error labels that occur most frequently in misclassifications of the four origin categories.**

| Frequent Mistaken labels | First Label | Second Label |
|---|---|---|
| *30 SPEED LIMIT* | 70 (57 times) | 60 (22 times) |
| *60 SPEED LIMIT* | 70 (82 times) | 80 (8 times) |
| *90 SPEED LIMIT* | 70 (69 times) | 100 (9 times) |
| *STOP* | 30 (67 times) | 60 (14 times) |

## 5 CONCLUSION

In this paper, we proposed a dynamic robust laser attack framework combined with embodied intelligence: ELA, which contained three stages: perception, decision and control to achieve the attack. For the perception stage, a PTN was designed to infer key local information in the victim's view without direct interaction with its sensor. For the decision and control stage, an attack decision-making agent was designed and trained with reinforcement learning, which could timely generate proper strategies according to scene variations and then implement them flexibly. This is a new paradigm to enhance the robustness of contactless attacks better suitable to real-world successive scenarios, which may pave a new path for enhancing the robustness of physical attacks.

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
