# OpenReview forum: "Embodied Laser Attack:Leveraging Scene Priors to Achieve Agent-based Robust Non-contact Attacks"
_acmmm.org/ACMMM/2024/Conference — MM2024 Poster_

### Official Review · Reviewer_WtL4 · 2024-05-23

**Rating:** 3
**Confidence:** 3

**Summary:**

The paper presents a novel embodied laser attack method that combines perception, decision-making, and control modules to achieve efficient and robust non-contact physical attacks in dynamic scenes. This study not only enhances the practicality and concealment of physical adversarial attacks but also provides new ideas for future research on physical attacks.

**Strengths:**

- The paper proposes an innovative non-contact laser attack framework (ELA), which combines the Perception-Decision-Control paradigm of embodied intelligence, demonstrating enhanced robustness of laser attacks in dynamic scenarios.

- The effectiveness of the ELA framework is validated through experiments in the CARLA simulator, with performance comparisons against different methods under dynamic scenarios.

-  The paper includes detailed comparisons with existing methods (such as EOT), highlighting the higher attack success rate and real-time performance of ELA in dynamic continuous scenarios.

**Limitations:**

- Although the framework has been validated in a simulator, there is limited verification in real physical environments. The paper only conducts preliminary simulations and lacks discussion on complex situations that might arise in real-world applications. This affects the reliability and applicability of the method in practical scenarios.

-  While the advantages of ELA are showcased, there is insufficient discussion on its limitations. For instance, how does the method perform under different weather and lighting conditions? Is it still effective with more diverse traffic signs and more complex road environments?

-  The method has a high dependency on training data and models, but the paper does not provide detailed explanations on how different data and models impact attack effectiveness. Do different traffic scenarios, camera positions, and vehicle speeds significantly affect the attack performance?

- The paper focuses entirely on vision-related topics rather than covering multimedia aspects.

**Suitability:**

1

---

### Official Review · Reviewer_Qa6d · 2024-05-25

**Rating:** 3
**Confidence:** 3

**Summary:**

The paper presents a novel framework for conducting non-contact adversarial attacks in dynamic environments such as traffic scenarios. This framework, referred to as Embodied Laser Attack (ELA), integrates the embodied intelligence paradigm, combining perception, decision, and control processes to adaptively generate adversarial attacks using a controllable laser system. It utilizes a data-driven reinforcement learning approach for dynamic decision-making, enabling the system to react instantaneously to changes in the environment. The paper demonstrates the effectiveness of this approach through extensive experiments in simulated traffic scenarios, highlighting the potential for real-world applications.

**Strengths:**

1. The paper introduces a groundbreaking approach by integrating the embodied intelligence framework, which effectively combines perception, decision-making, and control into a cohesive system for non-contact adversarial attacks.

2. By leveraging intrinsic geometric priors specific to traffic scenarios, the proposed Embodied Laser Attack (ELA) method significantly reduces the computational complexity typically associated with perspective transformations.

3. The framework employs a reinforcement learning algorithm to train an attack agent that can make real-time decisions based on the perceived state of the environment. This method stands out for its ability to generate effective and immediate adversarial actions, a critical feature for scenarios where attack opportunities might be fleeting and highly context-dependent.

**Limitations:**

1. The paper primarily compares its ELA methodology with EOT, which is not specifically tailored for the specified application context of non-contact adversarial attacks.

2. were trained and implemented. This omission makes it difficult to ascertain the fairness and relevance of the comparisons made.

3. No ablation studies are provieded involving parameters such as $\gamma_1$ and $\gamma_2$, which are crucial for understanding the contribution of different components of the reward function.

**Suitability:**

2

---

### Official Review · Reviewer_7657 · 2024-05-27

**Rating:** 4
**Confidence:** 3

**Summary:**

This paper introduces the Embodied Laser Attack (ELA) framework, addressing the limitations of traditional physical adversarial attacks in dynamic scenarios like traffic situations. ELA utilizes embodied intelligence to dynamically tailor non-contact laser attacks, enhancing robustness through scene priors, reinforcement learning, and controllable laser emissions. The framework demonstrates effectiveness in various traffic scenarios, offering active adaptation for improved attack performance compared to static methods. The authors discuss challenges in victim perspective modeling and decision-making, highlighting ELA's innovative approach to enhancing non-contact adversarial attacks.

**Strengths:**

This paper presents a novel Embodied Laser Attack (ELA) framework that addresses the challenge of enhancing the robustness of non-contact adversarial attacks in dynamic scenarios like traffic situations. The strength of this paper lies in its innovative approach of leveraging embodied intelligence to dynamically tailor laser attacks, incorporating perception, decision-making, and control modules for real-time adaptation. By introducing ELA as an active adaptation method compared to traditional static methods like Expectation over Transformation (EOT), the paper showcases a shift toward flexible and efficient adversarial attacks in dynamic environments. Through rigorous experiments in both digital and physical environments, the paper demonstrates the effectiveness of ELA in various traffic scenarios, highlighting its potential to significantly contribute to the field of physical adversarial attacks. The systematic verification comparing ELA with EOT further reinforces the paper's strength in providing a comprehensive evaluation of the proposed framework.

**Limitations:**

This paper has the following limitations:
1. The authors claim that their ELA attack could leverage the embodied intelligence paradigm of perception-decision-control. However, the ELA method only attacks the perception module of the autopilot system. The relationship between ELA and embodied intelligence is unclear. Although the authors design the decision-control module, it is more like an optimizing strategy rather than attacking the control system in the embodied intelligence system. The authors should emphasize the relationship between ELA and the embodied intelligence paradigm.
2. The authors claim that ELA could make real-time decisions according to scenario changes in a dynamic manner. In Table 2, the authors claim the time cost is around 0.2s, which could attack the realistic system. However, the details of the time cost are unclear. Does it cover the whole perception-decision-control paradigm? If so, it would be better if the authors provided more details of the hardware, e.g., the processor number of CPU and GPU.
3. Although ELA, AdvLB, and AdvLS have proved their effectiveness on image classifiers, most of the real-world autonomous systems are based on object detectors. To inspire future research on laser attacks, the authors could provide some attempts on the detectors such as YOLO and Faster-RCNN.

**Suitability:**

2

---

### Official Review · Reviewer_1nTY · 2024-06-02

**Rating:** 4
**Confidence:** 3

**Summary:**

This paper proposes the first dynamic robust non-contact attack framework, which utilizes the paradigm of embodied intelligence: Perception-Decision-Control to dynamically and timely adjust a manipulable laser towards valid attack strategies according to current situations.

A novel Perspective Transformation Network(PTN) is proposed to enable rapid simulation of object variations relying on the intrinsic geometric priors of traffic scenes; An adversarial laser decision making agent is designed to facilitate real-time valid strategies for dynamic successive processes.

**Strengths:**

This paper proposes the first dynamic robust non-contact attack framework, which is a novel work in traffic scenes. Especially, the dynamic application demonstrated in the supplementary video, is much different from the previous works  AdvLB and AdvLS,
The performance of the proposed method is much greater compared with the similar existing works AdvLB and AdvLS.

**Limitations:**

The authors use many long sentences with adverbial and subordinate clauses, making it a little hard to read.
x^𝑡_{min}, y^𝑡_{min} should be (x^𝑡_{min}, y^𝑡_{min}), the same as x^𝑡_{max}, y^𝑡_{miax}.
The authors focus on the colors that human eyes can clearly perceived, ranging from 𝝀_{min}=400nm∼𝝀_{max}=700. But in the implementation scenario, the images/videos are taken by the cameras, is the range of colors still appropriate?
In the experiments, the sample dataset is not versatile, with only the signs “30”, “60”, “90”, and “stop”.
Besides, the scenario of the physical implementation is not suitable enough for the traffic, while the digital simulation is a traffic situation.

**Suitability:**

2

---

### Meta-Review · Area_Chair_yZnr · 2024-07-02

**Recommendation:** Accept (Poster)
**Confidence:** 4

**Metareview:**

The paper got three positive scores and one negative score, and the average score is above the borderline. I would like to accept this paper.